# GNNExplainer: Generating Explanations for Graph Neural Networks

Rex Ying[†]     Dylan Bourgeois[†,‡]     Jiaxuan You[†]     Marinka Zitnik[†]     Jure Leskovec[†]

[†]Department of Computer Science, Stanford University
[‡]Robust.AI
{rexying, dtsbourg, jiaxuan, marinka, jure}@cs.stanford.edu

## Abstract

Graph Neural Networks (GNNs) are a powerful tool for machine learning on graphs. GNNs combine node feature information with the graph structure by recursively passing neural messages along edges of the input graph. However, incorporating both graph structure and feature information leads to complex models and explaining predictions made by GNNs remains unsolved. Here we propose GNNEXPLAINER, the first general, model-agnostic approach for providing interpretable explanations for predictions of any GNN-based model on any graph-based machine learning task. Given an instance, GNNEXPLAINER identifies a compact subgraph structure and a small subset of node features that have a crucial role in GNN's prediction. Further, GNNEXPLAINER can generate consistent and concise explanations for an entire class of instances. We formulate GNNEXPLAINER as an optimization task that maximizes the mutual information between a GNN's prediction and distribution of possible subgraph structures. Experiments on synthetic and real-world graphs show that our approach can identify important graph structures as well as node features, and outperforms alternative baseline approaches by up to 43.0% in explanation accuracy. GNNEXPLAINER provides a variety of benefits, from the ability to visualize semantically relevant structures to interpretability, to giving insights into errors of faulty GNNs.

## 1 Introduction

In many real-world applications, including social, information, chemical, and biological domains, data can be naturally modeled as graphs [9, 41, 49]. Graphs are powerful data representations but are challenging to work with because they require modeling of rich relational information as well as node feature information [45, 46]. To address this challenge, Graph Neural Networks (GNNs) have emerged as state-of-the-art for machine learning on graphs, due to their ability to recursively incorporate information from neighboring nodes in the graph, naturally capturing both graph structure and node features [16, 21, 40, 44].

Despite their strengths, GNNs lack transparency as they do not easily allow for a human-intelligible explanation of their predictions. Yet, the ability to understand GNN's predictions is important and useful for several reasons: (i) it can increase trust in the GNN model, (ii) it improves model's transparency in a growing number of decision-critical applications pertaining to fairness, privacy and other safety challenges [11], and (iii) it allows practitioners to get an understanding of the network characteristics, identify and correct systematic patterns of mistakes made by models before deploying them in the real world.

While currently there are no methods for explaining GNNs, recent approaches for explaining other types of neural networks have taken one of two main routes. One line of work locally approximates

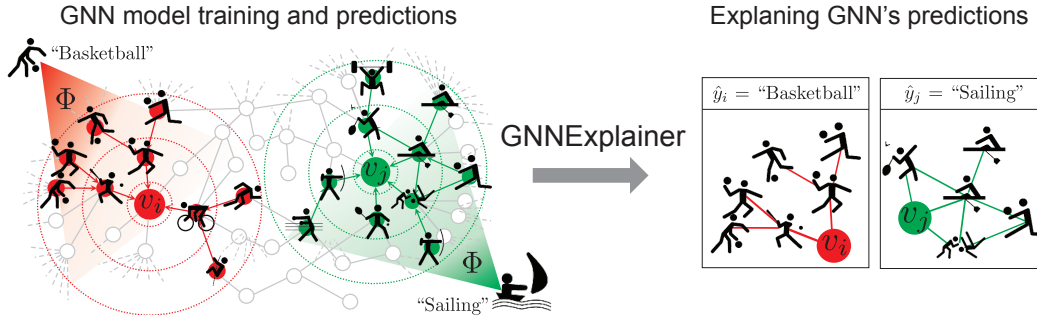

Figure 1: GNNEXPLAINER provides interpretable explanations for predictions made by any GNN model on any graph-based machine learning task. Shown is a hypothetical node classification task where a GNN model $\Phi$ is trained on a social interaction graph to predict future sport activities. Given a trained GNN $\Phi$ and a prediction $\hat{y}_i$ = "Basketball" for person $v_i$, GNNEXPLAINER generates an explanation by identifying a small subgraph of the input graph together with a small subset of node features (shown on the right) that are most influential for $\hat{y}_i$. Examining explanation for $\hat{y}_i$, we see that many friends in one part of $v_i$'s social circle enjoy ball games, and so the GNN predicts that $v_i$ will like basketball. Similarly, examining explanation for $\hat{y}_j$, we see that $v_j$'s friends and friends of his friends enjoy water and beach sports, and so the GNN predicts $\hat{y}_j$ = "Sailing."

models with simpler surrogate models, which are then probed for explanations [25, 29, 30]. Other methods carefully examine models for relevant features and find good qualitative interpretations of high level features [6, 13, 27, 32] or identify influential input instances [23, 38]. However, these approaches fall short in their ability to incorporate relational information, the essence of graphs. Since this aspect is crucial for the success of machine learning on graphs, any explanation of GNN's predictions should leverage rich relational information provided by the graph as well as node features.

Here we propose GNNEXPLAINER, an approach for explaining predictions made by GNNs. GNNEX-PLAINER takes a trained GNN and its prediction(s), and it returns an explanation in the form of a small subgraph of the input graph together with a small subset of node features that are most influential for the prediction(s) (Figure 1). The approach is model-agnostic and can explain predictions of any GNN on any machine learning task for graphs, including node classification, link prediction, and graph classification. It handles single- as well as multi-instance explanations. In the case of single-instance explanations, GNNEXPLAINER explains a GNN's prediction for one particular instance (*i.e.*, a node label, a new link, a graph-level label). In the case of multi-instance explanations, GNNEXPLAINER provides an explanation that consistently explains a set of instances (*e.g.*, nodes of a given class).

GNNEXPLAINER specifies an explanation as a rich subgraph of the entire graph the GNN was trained on, such that the subgraph maximizes the mutual information with GNN's prediction(s). This is achieved by formulating a mean field variational approximation and learning a real-valued *graph mask* which selects the important subgraph of the GNN's computation graph. Simultaneously, GNNEXPLAINER also learns a *feature mask* that masks out unimportant node features (Figure 1).

We evaluate GNNEXPLAINER on synthetic as well as real-world graphs. Experiments show that GNNEXPLAINER provides consistent and concise explanations of GNN's predictions. On synthetic graphs with planted network motifs, which play a role in determining node labels, we show that GNNEXPLAINER accurately identifies the subgraphs/motifs as well as node features that determine node labels outperforming alternative baseline approaches by up to 43.0% in explanation accuracy. Further, using two real-world datasets we show how GNNEXPLAINER can provide important domain insights by robustly identifying important graph structures and node features that influence a GNN's predictions. Specifically, using molecular graphs and social interaction networks, we show that GNNEXPLAINER can identify important domain-specific graph structures, such as $NO_2$ chemical groups or ring structures in molecules, and star structures in Reddit threads. Overall, experiments demonstrate that GNNEXPLAINER provides consistent and concise explanations for GNN-based models for different machine learning tasks on graphs.

## 2 Related work

Although the problem of explaining GNNs is not well-studied, the related problems of interpretability and neural debugging received substantial attention in machine learning. At a high level, we can group those interpretability methods for non-graph neural networks into two main families.

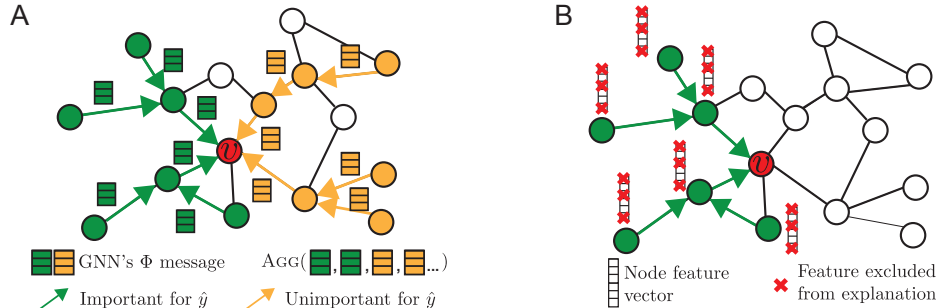

Figure 2: **A.** GNN computation graph $G_c$ (green and orange) for making prediction $\hat{y}$ at node $v$. Some edges in $G_c$ form important neural message-passing pathways (green), which allow useful node information to be propagated across $G_c$ and aggregated at $v$ for prediction, while other edges do not (orange). However, GNN needs to aggregate important as well as unimportant messages to form a prediction at node $v$, which can dilute the signal accumulated from $v$'s neighborhood. The goal of GNNEXPLAINER is to identify a small set of important features and pathways (green) that are crucial for prediction. **B.** In addition to $G_S$ (green), GNNEXPLAINER identifies what feature dimensions of $G_S$'s nodes are important for prediction by learning a node feature mask.

Methods in the first family formulate simple proxy models of full neural networks. This can be done in a model-agnostic way, usually by learning a locally faithful approximation around the prediction, for example through linear models [29] or sets of rules, representing sufficient conditions on the prediction [3, 25, 47]. Methods in the second family identify important aspects of the computation, for example, through feature gradients [13, 43], backpropagation of neurons' contributions to the input features [6, 31, 32], and counterfactual reasoning [19]. However, the saliency maps [43] produced by these methods have been shown to be misleading in some instances [2] and prone to issues like gradient saturation [31, 32]. These issues are exacerbated on discrete inputs such as graph adjacency matrices since the gradient values can be very large but only on very small intervals. Because of that, such approaches are not suitable for explaining predictions made by neural networks on graphs.

Instead of creating new, inherently interpretable models, post-hoc interpretability methods [1, 14, 15, 17, 23, 38] consider models as black boxes and then probe them for relevant information. However, no work has been done to leverage relational structures like graphs. The lack of methods for explaining predictions on graph-structured data is problematic, as in many cases, predictions on graphs are induced by a complex combination of nodes and paths of edges between them. For example, in some tasks, an edge is important only when another alternative path exists in the graph to form a cycle, and those two features, only when considered together, can accurately predict node labels [10, 12]. Their joint contribution thus cannot be modeled as a simple linear combinations of individual contributions.

Finally, recent GNN models augment interpretability via attention mechanisms [28, 33, 34]. However, although the learned edge attention values can indicate important graph structure, the values are the same for predictions across all nodes. Thus, this contradicts with many applications where an edge is essential for predicting the label of one node but not the label of another node. Furthermore, these approaches are either limited to specific GNN architectures or cannot explain predictions by jointly considering both graph structure and node feature information.

## 3 Formulating explanations for graph neural networks

Let $G$ denote a graph on edges $E$ and nodes $V$ that are associated with $d$-dimensional node features $\mathcal{X} = \{x_1, \ldots, x_n\}$, $x_i \in \mathbb{R}^d$. Without loss of generality, we consider the problem of explaining a node classification task (see Section 4.4 for other tasks). Let $f$ denote a label function on nodes $f : V \mapsto \{1, \ldots, C\}$ that maps every node in $V$ to one of $C$ classes. The GNN model $\Phi$ is optimized on all nodes in the training set and is then used for prediction, *i.e.*, to approximate $f$ on new nodes.

### 3.1 Background on graph neural networks

At layer $l$, the update of GNN model $\Phi$ involves three key computations [4, 45, 46]. (1) First, the model computes neural messages between every pair of nodes. The message for node pair $(v_i, v_j)$ is a function MSG of $v_i$'s and $v_j$'s representations $\mathbf{h}_i^{l-1}$ and $\mathbf{h}_j^{l-1}$ in the previous layer and of the relation $r_{ij}$ between the nodes: $m_{ij}^l = \text{MSG}(\mathbf{h}_i^{l-1}, \mathbf{h}_j^{l-1}, r_{ij})$. (2) Second, for each node $v_i$, GNN aggregates

messages from $v_i$'s neighborhood $\mathcal{N}_{v_i}$ and calculates an aggregated message $M_i$ via an aggregation method AGG [16, 35]: $M_i^l = \text{AGG}(\{m_{ij}^l | v_j \in \mathcal{N}_{v_i}\})$, where $\mathcal{N}_{v_i}$ is neighborhood of node $v_i$ whose definition depends on a particular GNN variant. (3) Finally, GNN takes the aggregated message $M_i^l$ along with $v_i$'s representation $\mathbf{h}_i^{l-1}$ from the previous layer, and it non-linearly transforms them to obtain $v_i$'s representation $\mathbf{h}_i^l$ at layer $l$: $\mathbf{h}_i^l = \text{UPDATE}(M_i^l, \mathbf{h}_i^{l-1})$. The final embedding for node $v_i$ after $L$ layers of computation is $\mathbf{z}_i = \mathbf{h}_i^L$. Our GNNEXPLAINER provides explanations for any GNN that can be formulated in terms of MSG, AGG, and UPDATE computations.

### 3.2  GNNEXPLAINER: Problem formulation

Our key insight is the observation that the computation graph of node $v$, which is defined by the GNN's neighborhood-based aggregation (Figure 2), fully determines all the information the GNN uses to generate prediction $\hat{y}$ at node $v$. In particular, $v$'s computation graph tells the GNN how to generate $v$'s embedding $\mathbf{z}$. Let us denote that computation graph by $G_c(v)$, the associated binary adjacency matrix by $A_c(v) \in \{0,1\}^{n \times n}$, and the associated feature set by $X_c(v) = \{x_j | v_j \in G_c(v)\}$. The GNN model $\Phi$ learns a conditional distribution $P_\Phi(Y | G_c, X_c)$, where $Y$ is a random variable representing labels $\{1, \dots, C\}$, indicating the probability of nodes belonging to each of $C$ classes.

A GNN's prediction is given by $\hat{y} = \Phi(G_c(v), X_c(v))$, meaning that it is fully determined by the model $\Phi$, graph structural information $G_c(v)$, and node feature information $X_c(v)$. In effect, this observation implies that we only need to consider graph structure $G_c(v)$ and node features $X_c(v)$ to explain $\hat{y}$ (Figure 2A). Formally, GNNEXPLAINER generates explanation for prediction $\hat{y}$ as $(G_S, X_S^F)$, where $G_S$ is a small subgraph of the computation graph. $X_S$ is the associated feature of $G_S$, and $X_S^F$ is a small subset of node features (masked out by the mask $F$, *i.e.*, $X_S^F = \{x_j^F | v_j \in G_S\}$) that are most important for explaining $\hat{y}$ (Figure 2B).

## 4  GNNEXPLAINER

Next we describe our approach GNNEXPLAINER. Given a trained GNN model $\Phi$ and a prediction (*i.e.*, single-instance explanation, Sections 4.1 and 4.2) or a set of predictions (*i.e.*, multi-instance explanations, Section 4.3), the GNNEXPLAINER will generate an explanation by identifying a subgraph of the computation graph and a subset of node features that are most influential for the model $\Phi$'s prediction. In the case of explaining a set of predictions, GNNEXPLAINER will aggregate individual explanations in the set and automatically summarize it with a prototype. We conclude this section with a discussion on how GNNEXPLAINER can be used for any machine learning task on graphs, including link prediction and graph classification (Section 4.4).

### 4.1  Single-instance explanations

Given a node $v$, our goal is to identify a subgraph $G_S \subseteq G_c$ and the associated features $X_S = \{x_j | v_j \in G_S\}$ that are important for the GNN's prediction $\hat{y}$. For now, we assume that $X_S$ is a small subset of $d$-dimensional node features; we will later discuss how to automatically determine which dimensions of node features need to be included in explanations (Section 4.2). We formalize the notion of importance using mutual information $MI$ and formulate the GNNEXPLAINER as the following optimization framework:

$$\max_{G_S} MI\left(Y, (G_S, X_S)\right) = H(Y) - H(Y | G = G_S, X = X_S). \tag{1}$$

For node $v$, $MI$ quantifies the change in the probability of prediction $\hat{y} = \Phi(G_c, X_c)$ when $v$'s computation graph is limited to explanation subgraph $G_S$ and its node features are limited to $X_S$.

For example, consider the situation where $v_j \in G_c(v_i)$, $v_j \neq v_i$. Then, if removing $v_j$ from $G_c(v_i)$ strongly decreases the probability of prediction $\hat{y}_i$, the node $v_j$ is a good counterfactual explanation for the prediction at $v_i$. Similarly, consider the situation where $(v_j, v_k) \in G_c(v_i)$, $v_j, v_k \neq v_i$. Then, if removing an edge between $v_j$ and $v_k$ strongly decreases the probability of prediction $\hat{y}_i$ then the absence of that edge is a good counterfactual explanation for the prediction at $v_i$.

Examining Eq. (1), we see that the entropy term $H(Y)$ is constant because $\Phi$ is fixed for a trained GNN. As a result, maximizing mutual information between the predicted label distribution $Y$ and

explanation $(G_S, X_S)$ is equivalent to minimizing conditional entropy $H(Y|G = G_S, X = X_S)$, which can be expressed as follows:

$$H(Y|G=G_S, X=X_S) = -\mathbb{E}_{Y|G_S, X_S}\left[\log P_\Phi(Y|G=G_S, X=X_S)\right]. \tag{2}$$

Explanation for prediction $\hat{y}$ is thus a subgraph $G_S$ that minimizes uncertainty of $\Phi$ when the GNN computation is limited to $G_S$. In effect, $G_S$ maximizes probability of $\hat{y}$ (Figure 2). To obtain a compact explanation, we impose a constraint on $G_S$'s size as: $|G_S| \leq K_M$, so that $G_S$ has at most $K_M$ nodes. In effect, this implies that GNNEXPLAINER aims to denoise $G_c$ by taking $K_M$ edges that give the highest mutual information with the prediction.

**GNNEXPLAINER's optimization framework.** Direct optimization of GNNEXPLAINER's objective is not tractable as $G_c$ has exponentially many subgraphs $G_S$ that are candidate explanations for $\hat{y}$. We thus consider a fractional adjacency matrix[1] for subgraphs $G_S$, *i.e.*, $A_S \in [0,1]^{n \times n}$, and enforce the subgraph constraint as: $A_S[j,k] \leq A_c[j,k]$ for all $j, k$. This continuous relaxation can be interpreted as a variational approximation of distribution of subgraphs of $G_c$. In particular, if we treat $G_S \sim \mathcal{G}$ as a random graph variable, the objective in Eq. (2) becomes:

$$\min_{\mathcal{G}} \mathbb{E}_{G_S \sim \mathcal{G}} H(Y|G = G_S, X = X_S), \tag{3}$$

With convexity assumption, Jensen's inequality gives the following upper bound:

$$\min_{\mathcal{G}} H(Y|G = \mathbb{E}_{\mathcal{G}}[G_S], X = X_S). \tag{4}$$

In practice, due to the complexity of neural networks, the convexity assumption does not hold. However, experimentally, we found that minimizing this objective with regularization often leads to a local minimum corresponding to high-quality explanations.

To tractably estimate $\mathbb{E}_{\mathcal{G}}$, we use mean-field variational approximation and decompose $\mathcal{G}$ into a multivariate Bernoulli distribution as: $P_{\mathcal{G}}(G_S) = \prod_{(j,k) \in G_c} A_S[j,k]$. This allows us to estimate the expectation with respect to the mean-field approximation, thereby obtaining $A_S$ in which $(j,k)$-th entry represents the expectation on whether edge $(v_j, v_k)$ exists. We observed empirically that this approximation together with a regularizer for promoting discreteness [40] converges to good local minima despite the non-convexity of GNNs. The conditional entropy in Equation 4 can be optimized by replacing the $\mathbb{E}_{\mathcal{G}}[G_S]$ to be optimized by a masking of the computation graph of adjacency matrix, $A_c \odot \sigma(M)$, where $M \in \mathbb{R}^{n \times n}$ denotes the mask that we need to learn, $\odot$ denotes element-wise multiplication, and $\sigma$ denotes the sigmoid that maps the mask to $[0,1]^{n \times n}$.

In some applications, instead of finding an explanation in terms of model's confidence, the users care more about "why does the trained model predict a certain class label", or "how to make the trained model predict a desired class label". We can modify the conditional entropy objective in Equation 4 with a cross entropy objective between the label class and the model prediction[2]. To answer these queries, a computationally efficient version of GNNEXPLAINER's objective, which we optimize using gradient descent, is as follows:

$$\min_{M} - \sum_{c=1}^{C} \mathbb{1}[y = c] \log P_\Phi(Y = y|G = A_c \odot \sigma(M), X = X_c), \tag{5}$$

The masking approach is also found in Neural Relational Inference [22], albeit with different motivation and objective. Lastly, we compute the element-wise multiplication of $\sigma(M)$ and $A_c$ and remove low values in $M$ through thresholding to arrive at the explanation $G_S$ for the GNN model's prediction $\hat{y}$ at node $v$.

## 4.2 Joint learning of graph structural and node feature information

To identify what node features are most important for prediction $\hat{y}$, GNNEXPLAINER learns a feature selector $F$ for nodes in explanation $G_S$. Instead of defining $X_S$ to consists of all node features, *i.e.*,

$X_S = \{x_j | v_j \in G_S\}$, GNNEXPLAINER considers $X_S^F$ as a subset of features of nodes in $G_S$, which are defined through a binary feature selector $F \in \{0, 1\}^d$ (Figure 2B):

$$X_S^F = \{x_j^F | v_j \in G_S\}, \quad x_j^F = [x_{j,t_1}, \dots, x_{j,t_k}] \text{ for } F_{t_i} = 1, \tag{6}$$

where $x_j^F$ has node features that are not masked out by $F$. Explanation $(G_S, X_S)$ is then jointly optimized for maximizing the mutual information objective:

$$\max_{G_S, F} MI\left(Y, (G_S, F)\right) = H(Y) - H(Y | G = G_S, X = X_S^F), \tag{7}$$

which represents a modified objective function from Eq. (1) that considers structural and node feature information to generate an explanation for prediction $\hat{y}$.

**Learning binary feature selector $F$.** We specify $X_S^F$ as $X_S \odot F$, where $F$ acts as a feature mask that we need to learn. Intuitively, if a particular feature is not important, the corresponding weights in GNN's weight matrix take values close to zero. In effect, this implies that masking the feature out does not decrease predicted probability for $\hat{y}$. Conversely, if the feature is important then masking it out would decrease predicted probability. However, in some cases this approach ignores features that are important for prediction but take values close to zero. To address this issue we marginalize over all feature subsets and use a Monte Carlo estimate to sample from empirical marginal distribution for nodes in $X_S$ during training [48]. Further, we use a reparametrization trick [20] to backpropagate gradients in Eq. (7) to the feature mask $F$. In particular, to backpropagate through a $d$-dimensional random variable $X$ we reparametrize $X$ as: $X = Z + (X_S - Z) \odot F$ s.t. $\sum_j F_j \leq K_F$, where $Z$ is a $d$-dimensional random variable sampled from the empirical distribution and $K_F$ is a parameter representing the maximum number of features to be kept in the explanation.

**Integrating additional constraints into explanations.** To impose further properties on the explanation we can extend GNNEXPLAINER's objective function in Eq. (7) with regularization terms. For example, we use element-wise entropy to encourage structural and node feature masks to be discrete. Further, GNNEXPLAINER can encode domain-specific constraints through techniques like Lagrange multiplier of constraints or additional regularization terms. We include a number of regularization terms to produce explanations with desired properties. We penalize large size of the explanation by adding the sum of all elements of the mask paramters as the regularization term.

Finally, it is important to note that each explanation must be a valid computation graph. In particular, explanation $(G_S, X_S)$ needs to allow GNN's neural messages to flow towards node $v$ such that GNN can make prediction $\hat{y}$. Importantly, GNNEXPLAINER automatically provides explanations that represent valid computation graphs because it optimizes structural masks across entire computation graphs. Even if a disconnected edge is important for neural message-passing, it will not be selected for explanation as it cannot influence GNN's prediction. In effect, this implies that the explanation $G_S$ tends to be a small connected subgraph.

### 4.3 Multi-instance explanations through graph prototypes

The output of a single-instance explanation (Sections 4.1 and 4.2) is a small subgraph of the input graph and a small subset of associated node features that are most influential for a single prediction. To answer questions like "How did a GNN predict that a given set of nodes all have label $c$?", we need to obtain a global explanation of class $c$. Our goal here is to provide insight into how the identified subgraph for a particular node relates to a graph structure that explains an entire class. GNNEXPLAINER can provide multi-instance explanations based on graph alignments and prototypes. Our approach has two stages:

First, for a given class $c$ (or, any set of predictions that we want to explain), we first choose a reference node $v_c$, for example, by computing the mean embedding of all nodes assigned to $c$. We then take explanation $G_S(v_c)$ for reference $v_c$ and align it to explanations of other nodes assigned to class $c$. Finding optimal matching of large graphs is challenging in practice. However, the single-instance GNNEXPLAINER generates small graphs (Section 4.2) and thus near-optimal pairwise graph matchings can be efficiently computed.

Second, we aggregate aligned adjacency matrices into a graph prototype $A_{\text{proto}}$ using, for example, a robust median-based approach. Prototype $A_{\text{proto}}$ gives insights into graph patterns shared between nodes that belong to the same class. One can then study prediction for a particular node by comparing explanation for that node's prediction (*i.e.*, returned by single-instance explanation approach) to the prototype (see Appendix for more information).

### 4.4 GNNEXPLAINER model extensions

**Any machine learning task on graphs.** In addition to explaining node classification, GNNEX-PLAINER provides explanations for link prediction and graph classification with no change to its optimization algorithm. When predicting a link $(v_j, v_k)$, GNNEXPLAINER learns two masks $X_S(v_j)$ and $X_S(v_k)$ for both endpoints of the link. When classifying a graph, the adjacency matrix in Eq. (5) is the union of adjacency matrices for all nodes in the graph whose label we want to explain. However, note that in graph classification, unlike node classification, due to the aggregation of node embeddings, it is no longer true that the explanation $G_S$ is necessarily a connected subgraph. Depending on application, in some scenarios such as chemistry where explanation is a functional group and should be connected, one can extract the largest connected component as the explanation.

**Any GNN model.** Modern GNNs are based on message passing architectures on the input graph. The message passing computation graphs can be composed in many different ways and GNNEXPLAINER can account for all of them. Thus, GNNEXPLAINER can be applied to: Graph Convolutional Networks [21], Gated Graph Sequence Neural Networks [26], Jumping Knowledge Networks [36], Attention Networks [33], Graph Networks [4], GNNs with various node aggregation schemes [7, 5, 18, 16, 40, 39, 35], Line-Graph NNs [8], position-aware GNN [42], and many other GNN architectures.

**Computational complexity.** The number of parameters in GNNEXPLAINER's optimization depends on the size of computation graph $G_c$ for node $v$ whose prediction we aim to explain. In particular, $G_c(v)$'s adjacency matrix $A_c(v)$ is equal to the size of the mask $M$, which needs to be learned by GNNEXPLAINER. However, since computation graphs are typically relatively small, compared to the size of exhaustive $L$-hop neighborhoods (*e.g.*, 2-3 hop neighborhoods [21], sampling-based neighborhoods [39], neighborhoods with attention [33]), GNNEXPLAINER can effectively generate explanations even when input graphs are large.

## 5  Experiments

We begin by describing the graphs, alternative baseline approaches, and experimental setup. We then present experiments on explaining GNNs for node classification and graph classification tasks. Our qualitative and quantitative analysis demonstrates that GNNEXPLAINER is accurate and effective in identifying explanations, both in terms of graph structure and node features.

**Synthetic datasets.** We construct four kinds of node classification datasets (Table 1). (1) In BA-SHAPES, we start with a base Barabási-Albert (BA) graph on 300 nodes and a set of 80 five-node "house"-structured network motifs, which are attached to randomly selected nodes of the base graph. The resulting graph is further perturbed by adding $0.1N$ random edges. Nodes are assigned to 4 classes based on their structural roles. In a house-structured motif, there are 3 types of roles: the top, middle and bottom node of the house. Therefore there are 4 different classes, corresponding to nodes at the top, middle, bottom of houses, and nodes that do not belong to a house. (2) BA-COMMUNITY dataset is a union of two BA-SHAPES graphs. Nodes have normally distributed feature vectors and are assigned to one of 8 classes based on their structural roles and community memberships. (3) In TREE-CYCLES, we start with a base 8-level balanced binary tree and 80 six-node cycle motifs, which are attached to random nodes of the base graph. (4) TREE-GRID is the same as TREE-CYCLES except that 3-by-3 grid motifs are attached to the base tree graph in place of cycle motifs.

**Real-world datasets.** We consider two graph classification datasets: (1) MUTAG is a dataset of 4,337 molecule graphs labeled according to their mutagenic effect on the Gram-negative bacterium *S. typhimurium* [10]. (2) REDDIT-BINARY is a dataset of 2,000 graphs, each representing an online discussion thread on Reddit. In each graph, nodes are users participating in a thread, and edges indicate that one user replied to another user's comment. Graphs are labeled according to the type of user interactions in the thread: *r/IAmA* and *r/AskReddit* contain Question-Answer interactions, while *r/TrollXChromosomes* and *r/atheism* contain Online-Discussion interactions [37].

**Alternative baseline approaches.** Many explainability methods cannot be directly applied to graphs (Section 2). Nevertheless, we here consider the following alternative approaches that can provide insights into predictions made by GNNs: (1) GRAD is a gradient-based method. We compute gradient of the GNN's loss function with respect to the adjacency matrix and the associated node features, similar to a saliency map approach. (2) ATT is a graph attention GNN (GAT) [33] that learns attention weights for edges in the computation graph, which we use as a proxy measure of edge importance. While ATT does consider graph structure, it does not explain using node features and can only explain GAT models. Furthermore, in ATT it is not obvious which attention weights need to be used for edge

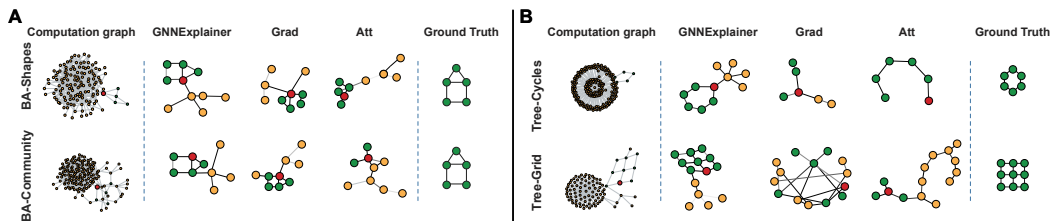

|  | BA-Shapes | BA-Community | Tree-Cycles | Tree-Grid |
|---|---|---|---|---|
| **Base** | | Community 0   Community 1 | | |
| **Motif** | | | | |
| **Node Features** | None | $\mathcal{N}(\mu_l, \sigma_l)$ where $l$ = community ID | None | None |
| **Explanation content** | Graph structure | Graph structure Node feature information | Graph structure | Graph structure |
| **Explanation accuracy** | | | | |
| Att | 0.815 | 0.739 | 0.824 | 0.612 |
| Grad | 0.882 | 0.750 | 0.905 | 0.667 |
| GNNExplainer | **0.925** | **0.836** | **0.948** | **0.875** |

Table 1: Illustration of synthetic datasets (refer to "Synthetic datasets" for details) together with performance evaluation of GNNEXPLAINER and alternative baseline explainability approaches.

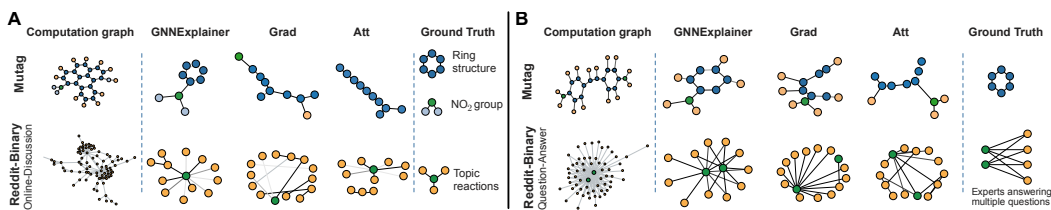

Figure 3: Evaluation of single-instance explanations. **A-B.** Shown are exemplar explanation subgraphs for node classification task on four synthetic datasets. Each method provides explanation for the red node's prediction.

importance, since a 1-hop neighbor of a node can also be a 2-hop neighbor of the same node due to cycles. Each edge's importance is thus computed as the average attention weight across all layers.

**Setup and implementation details.** For each dataset, we first train a single GNN for each dataset, and use GRAD and GNNEXPLAINER to explain the predictions made by the GNN. Note that the ATT baseline requires using a graph attention architecture like GAT [33]. We thus train a separate GAT model on the same dataset and use the learned edge attention weights for explanation. Hyperparameters $K_M, K_F$ control the size of subgraph and feature explanations respectively, which is informed by prior knowledge about the dataset. For synthetic datasets, we set $K_M$ to be the size of ground truth. On real-world datasets, we set $K_M = 10$. We set $K_F = 5$ for all datasets. We further fix our weight regularization hyperparameters across all node and graph classification experiments. We refer readers to the Appendix for more training details (Code and datasets are available at https://github.com/RexYing/gnn-model-explainer).

**Results.** We investigate questions: Does GNNEXPLAINER provide sensible explanations? How do explanations compare to the ground-truth knowledge? How does GNNEXPLAINER perform on various graph-based prediction tasks? Can it explain predictions made by different GNNs?

**1) Quantitative analyses.** Results on node classification datasets are shown in Table 1. We have ground-truth explanations for synthetic datasets and we use them to calculate explanation accuracy for all explanation methods. Specifically, we formalize the explanation problem as a binary classification task, where edges in the ground-truth explanation are treated as labels and importance weights given by explainability method are viewed as prediction scores. A better explainability method predicts

Figure 4: Evaluation of single-instance explanations. **A-B.** Shown are exemplar explanation subgraphs for graph classification task on two datasets, MUTAG and REDDIT-BINARY.

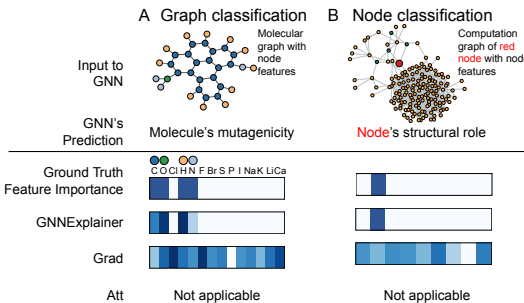

Figure 5: Visualization of features that are important for a GNN's prediction. **A.** Shown is a representative molecular graph from MUTAG dataset (top). Importance of the associated graph features is visualized with a heatmap (bottom). In contrast with baselines, GNNEXPLAINER correctly identifies features that are important for predicting the molecule's mutagenicity, *i.e.* C, O, H, and N atoms. **B.** Shown is a computation graph of a red node from BA-COMMUNITY dataset (top). Again, GNNEXPLAINER successfully identifies the node feature that is important for predicting the structural role of the node but baseline methods fail.

high scores for edges that are in the ground-truth explanation, and thus achieves higher explanation accuracy. Results show that GNNEXPLAINER outperforms alternative approaches by 17.1% on average. Further, GNNEXPLAINER achieves up to 43.0% higher accuracy on the hardest TREE-GRID dataset.

**2) Qualitative analyses.** Results are shown in Figures 3–5. In a topology-based prediction task with no node features, *e.g.* BA-SHAPES and TREE-CYCLES, GNNEXPLAINER correctly identifies network motifs that explain node labels, *i.e.* structural labels (Figure 3). As illustrated in the figures, house, cycle and tree motifs are identified by GNNEXPLAINER but not by baseline methods. In Figure 4, we investigate explanations for graph classification task. In MUTAG example, colors indicate node features, which represent atoms (hydrogen H, carbon C, *etc*). GNNEXPLAINER correctly identifies carbon ring as well as chemical groups $NH_2$ and $NO_2$, which are known to be mutagenic [10].

Further, in REDDIT-BINARY example, we see that Question-Answer graphs (2nd row in Figure 4B) have 2-3 high degree nodes that simultaneously connect to many low degree nodes, which makes sense because in QA threads on Reddit we typically have 2-3 experts who all answer many different questions [24]. Conversely, we observe that discussion patterns commonly exhibit tree-like patterns (2nd row in Figure 4A), since a thread on Reddit is usually a reaction to a single topic [24]. On the other hand, GRAD and ATT methods give incorrect or incomplete explanations. For example, both baseline methods miss cycle motifs in MUTAG dataset and more complex grid motifs in TREE-GRID dataset. Furthermore, although edge attention weights in ATT can be interpreted as importance scores for message passing, the weights are shared across all nodes in input the graph, and as such ATT fails to provide high quality single-instance explanations.

An essential criterion for explanations is that they must be interpretable, *i.e.*, provide a qualitative understanding of the relationship between the input nodes and the prediction. Such a requirement implies that explanations should be easy to understand while remaining exhaustive. This means that a GNN explainer should take into account both the structure of the underlying graph as well as the associated features when they are available. Figure 5 shows results of an experiment in which GNNEXPLAINER jointly considers structural information as well as information from a small number of feature dimensions[3]. While GNNEXPLAINER indeed highlights a compact feature representation in Figure 5, gradient-based approaches struggle to cope with the added noise, giving high importance scores to irrelevant feature dimensions.

# 6 Conclusion

We present GNNEXPLAINER, a novel method for explaining predictions of any GNN on any graph-based machine learning task without requiring modification of the underlying GNN architecture or re-training. We show how GNNEXPLAINER can leverage recursive neighborhood-aggregation scheme of graph neural networks to identify important graph pathways as well as highlight relevant node feature information that is passed along edges of the pathways. While the problem of explainability of machine-learning predictions has received substantial attention in recent literature, our work is unique in the sense that it presents an approach that operates on relational structures—graphs with rich node features—and provides a straightforward interface for making sense out of GNN predictions, debugging GNN models, and identifying systematic patterns of mistakes.

**Acknowledgments**

Jure Leskovec is a Chan Zuckerberg Biohub investigator. We gratefully acknowledge the support of DARPA under FA865018C7880 (ASED) and MSC; NIH under No. U54EB020405 (Mobilize); ARO under No. 38796-Z8424103 (MURI); IARPA under No. 2017-17071900005 (HFC), NSF under No. OAC-1835598 (CINES) and HDR; Stanford Data Science Initiative, Chan Zuckerberg Biohub, JD.com, Amazon, Boeing, Docomo, Huawei, Hitachi, Observe, Siemens, UST Global. The U.S. Government is authorized to reproduce and distribute reprints for Governmental purposes notwithstanding any copyright notation thereon. Any opinions, findings, and conclusions or recommendations expressed in this material are those of the authors and do not necessarily reflect the views, policies, or endorsements, either expressed or implied, of DARPA, NIH, ONR, or the U.S. Government.

## Footnotes

[1] For typed edges, we define $G_S \in [0,1]^{C_e \times n \times n}$ where $C_e$ is the number of edge types.

[2] The label class is the predicted label class by the GNN model to be explained, when answering "why does the trained model predict a certain class label". "how to make the trained model predict a desired class label" can be answered by using the ground-truth label class.

[3]Feature explanations are shown for the two datasets with node features, *i.e.*, MUTAG and BA-COMMUNITY.

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
