[Supplementary Material]

# Appendix for "GNNExplainer: Generating Explanations for Graph Neural Networks"

Rex Ying[†], Dylan Bourgeois[†,‡], Jiaxuan You[†], Marinka Zitnik[†], and Jure Leskovec[†]

[†]Department of Computer Science, Stanford University
[‡]Robust.AI
{rexying, dtsbourg, jiaxuan, marinka, jure}@cs.stanford.edu

## A  Multi-instance explanations

The problem of multi-instance explanations for graph neural networks is challenging and an important area to study.

Here we propose a solution based on GNNEXPLAINER to find common components of explanations for a set of 10 explanations for 10 different instances in the same label class. More research in this area is necessary to design efficient Multi-instance explanation methods. The main challenges in practice is mainly due to the difficulty to perform graph alignment under noise and variances of node neighborhood structures for nodes in the same class. The problem is closely related to finding the maximum common subgraphs of explanation graphs, which is an NP-hard problem. In the following we introduces a neural approach to this problem. However, note that existing graph libraries (based on heuristics or integer programming relaxation) to find the maximal common subgraph of graphs can be employed to replace the neural components of the following procedure, when trying to identify and align with a prototype.

The output of a single-instance GNNEXPLAINER indicates what graph structural and node feature information is important for a given prediction. To obtain an understanding of "why is a given set of nodes classified with label $y$", we want to also obtain a global explanation of the class, which can shed light on how the identified structure for a given node is related to a prototypical structure unique for its label. To this end, we propose an alignment-based multi-instance GNNEXPLAINER.

For any given class, we first choose a reference node. Intuitively, this node should be a prototypical node for the class. Such node can be found by computing the mean of the embeddings of all nodes in the class, and choose the node whose embedding is the closest to the mean. Alternatively, if one has prior knowledge about the important computation subgraph, one can choose one which matches most to the prior knowledge.

Given the reference node for class $c$, $v_c$, and its associated important computation subgraph $G_S(v_c)$, we align each of the identified computation subgraphs for all nodes in class $c$ to the reference $G_S(v_c)$. Utilizing the idea in the context of differentiable pooling [1], we use the a relaxed alignment matrix to find correspondence between nodes in an computation subgraph $G_S(v)$ and nodes in the reference computation subgraph $G_S(v_c)$. Let $A_v$ and $X_v$ be the adjacency matrix and the associated feature matrix of the to-be-aligned computation subgraph. Similarly let $A^*$ be the adjacency matrix and associated feature matrix of the reference computation subgraph. Then we optimize the relaxed alignment matrix $P \in \mathbb{R}^{n_v \times n^*}$, where $n_v$ is the number of nodes in $G_S(v)$, and $n^*$ is the number of nodes in $G_S(v_c)$ as follows:

$$\min_P |P^T A_v P - A^*| + |P^T X_v - X^*|. \tag{1}$$

Figure 1: GNNEXPLAINER is able to provide a prototype for a given node class, which can help identify functional subgraphs, e.g. a mutagenic compound from the MUTAG dataset.

The first term in Eq. (1) specifies that after alignment, the aligned adjacency for $G_S(v)$ should be as close to $A^*$ as possible. The second term in the equation specifies that the features should for the aligned nodes should also be close.

In practice, it is often non-trivial for the relaxed graph matching to find a good optimum for matching 2 large graphs. However, thanks to the single-instance explainer, which produces concise subgraphs for important message-passing, a matching that is close to the best alignment can be efficiently computed.

**Prototype by alignment.** We align the adjacency matrices of all nodes in class $c$, such that they are aligned with respect to the ordering defined by the reference adjacency matrix. We then use median to generate a prototype that is resistent to outliers, $A_{\mathrm{proto}} = \mathrm{median}(A_i)$, where $A_i$ is the aligned adjacency matrix representing explanation for $i$-th node in class $c$. Prototype $A_{\mathrm{proto}}$ allows users to gain insights into structural graph patterns shared between nodes that belong to the same class. Users can then investigate a particular node by comparing its explanation to the class prototype.

## B  Experiments on multi-instance explanations and prototypes

In the context of multi-instance explanations, an explainer must not only highlight information locally relevant to a particular prediction, but also help emphasize higher-level correlations across instances. These instances can be related in arbitrary ways, but the most evident is class-membership. The assumption is that members of a class share common characteristics, and the model should help highlight them. For example, mutagenic compounds are often found to have certain characteristic functional groups that such $NO_2$, a pair of Oxygen atoms together with a Nitrogen atom. A trained eye might notice that Figure 1 already hints at their presence. The evidence grows stronger when a prototype is generated by GNNEXPLAINER, shown in Figure 1. The model is able to pick-up on this functional structure, and promote it as archetypal of mutagenic compounds.

## C  Further implementation details

**Training details.** We use the Adam optimizer to train both the GNN and explaination methods. All GNN models are trained for 1000 epochs with learning rate 0.001, reaching accuracy of at least 85% for graph classification datasets, and 95% for node classification datasets. The train/validation/test split is $80/10/10\%$ for all datasets. In GNNEXPLAINER, we use the same optimizer and learning rate, and train for 100 - 300 epochs. This is efficient since GNNEXPLAINER only needs to be trained on a local computation graph with $< 100$ nodes.

**Regularization.** In addition to graph size constraint and graph laplacian constraint, we further impose the feature size constraint, which constrains that the number of unmasked features do not exceed a threshold. The regularization hyperparameters for subgraph size is 0.005; for laplacian is 0.5; for feature explanation is 0.1. The same values of hyperparameters are used across all experiments.

**Subgraph extraction.** To extract the explanation subgraph $G_S$, we first compute the importance weights on edges (gradients for GRAD baseline, attention weights for ATT baseline, and masked adjacency for GNNEXPLAINER). A threshold is used to remove low-weight edges, and identify the explanation subgraph $G_S$. The ground truth explanations of all datasets are connected subgraphs. Therefore, we identify the explanation as the connected component containing the explained node in $G_S$. For graph classification, we identify the explanation by the maximum connected component of $G_S$. For all methods, we perform a search to find the maximum threshold such that the explanation is

at least of size $K_M$. When multiple edges have tied importance weights, all of them are included in the explanation.

## References

[1] Z. Ying, J. You, C. Morris, X. Ren, W. Hamilton, and J. Leskovec. Hierarchical graph representation learning with differentiable pooling. In *NeurIPS*, 2018.