[Reviews · NeurIPS 2019]

Reviewer 1



originality: graph neural networks did not have a method for explainable predictions significance: GNNs are widely used and explanations are important. Work shows existing NN explanation frameworks don't work well with graphs quality: The authors propose a novel method to explain the predictions of Graph Neural Networks - both for explaining single node predictions or predictions of a class of nodes. This is achieved by isolating multiple possible subgraphs relevant to the classification, whose edges are selected via mutual information maximization and variational inference. Use of masks allows for explanations to jointly include node features. Convincing empirical results (except for how the hand-picked thresholding affects the results). significance: The paper is solid and make good arguments for the decisions made in the method. some issues: 1. The proposal distribution for VI assumes independent edges in the subgraph, which is not ideal if one thinks that some explanations must jointly depend on multiple edges: say, either edges (a,b) and (b,c) or edges (a,d) and (d,c). 2. A threshold is used to remove low-weight edges, and identify the explanation subgraph G_S. It is unclear how the different explanation methods are affected by the choice of this arbitrary threshold. Moreover, there isn't justification for the chosen number of edges and nodes selected to justify the classification. (minor: Automatically discovering subgraph sizes would be a nice feature to have). 3. minor: The work does not provide an assessment of how well the mean field approximates the original objective. Minor typos: "is fixed, trained GNN" => " is fixed, a trained GNN"

Reviewer 2



The authors propose a model-agnostic approach for explaining GNNs prediction. The problem they are addressing is novel, and they use a simple but effective approach to address that. The paper is well written and most of the details are clear. Their approach is novel in finding important subgraphs for the prediction. They evaluate their method across mulitple synthetic and real world dataset. A more detailed explaination on synthetic dataset and how they generate the labels would make the experiments more reproducable. quantitative analysis on the real world datasets in addition to the synthetic datasets would be beneficial.

Reviewer 3



The paper focuses on graph neural networks (GNN) that has recently gained significant attention as vital components for machine learning systems over graph structured data. Specifically, the authors build a method to analyze the predictions made by GNN and output explanations in the form of contextual subgraph and subset of features of nodes in this subgraph. The goal is to understand what parts of the graph (structure and features) were given importance by GNN model while computing predictions for node and edge. The authors achieve this by optimizing an information theoretic objective that considers the mutual information between the predictions and the relevant graph components. The authors provide methods for both single and multi-instance setting. Empirical evidence show that GNNExplainer outperforms couple of baselines on various datasets in terms of explanation accuracy and identifying the correct subgraphs/subset of features. - Overall, the paper is well written and provides good exposition of the technical details of the approach. - Since past few years, there has been significant increase in machine learning over graphs while little effort has been spent on validating and/or interpreting the resulting models. This paper certainly sets a right step in this direction and is a very timely contribution - The idea of using structural properties such as subgraph and important features aligns well with what one expects the GNN models to focus on, while making predictions. Having said that, I have several concerns both for methodology but more importantly, for experiments that prevents me from giving high score to the paper: - It is not clear what is the strong motivation behind using mutual information? One can see that the mutual information is useful in connecting predictions to substructures however, is this guaranteed to be always useful? For instance, if two nodes are connected to each other within a dense graph but have different labels, how does this approach able to figure out good sub-component for one node different from the other node which leads to different labels? - On page 4, the authors mention K_M nodes and also K_M edges. is this a typo? If not how K_M node and K_M edges are related here? - Different constraints are a bit concerning in this approach: - The constraint of K_M edges seem to be very restrictive. It appears from experiments it is a hyper-parameter. So, when one set it to a small number, isn’t the explainer forced to give very naive explanations? e.g. mostly as subset of 1-ho neighbors? - Further, a lot of GNN models have been developed with a promise that one can capture information from far away nodes through message passing atleast in few hop (>1) neighborhood. But the restriction in the paper to have connected subgraph as explanation is counterintuitive to that. - What is the purpose of subgraph constraint A_S? And why is it explicitly required? Isn’t it naturally enforced based on definition? - Further, how do you ensure that these restrictions in-fact, are not contaminating the explanation i.e. the explanation obtained for the prediction are mainly the result of this restrictive approach and not the artifact of the model? - While the authors mention in a section that this approach is model-agnostic and can be applied to any model post-facto, it is still important to see how (if anything) would be required to change when intricacies of some other model are involved? Does it mean that for any other model, exact same objective will still be able to lead to accurate explanations? Or will you need to change the objective (input to the objective ,e.g. what to condition the predictions on), based on the model? - If a GNN model uses simple attention mechanism on features (to distinguish from GAT), will GNN Explainer still perform able to capture important features effectively? Experiments: —————- - How much role did regularization play in the overall performance? You have multiple components in the. loss function, can you provide an ablation on what happens if you only optimize for structure vs feature vs both (plus regularization ,etc)? - For the quantitative analysis on synthetic data, how do you come up with the ground truth explanation? Is it curated manually? - Overall, the experiments are very limited and does not fully expose the use of this approach: - Specifically, the authors mention multiple times that such a tool can be used to analyse false predictions made by model. It is very important for this paper to provide empirical (quantitative or qualitative) to this with some examples may be where the model does bad prediction and the explanation is able to capture it. - Further, while the authors compare with few weak baselines (this is not a negative, it is understandable that no strong baseline exist) on multiple datasets, a more insightful experiment would be to compare different GNN models on same dataset and demonstrate how the GNNExplainer captures different characteristics for different models. - While there are some experiments shown for subgraph based explanation, experiments on node features are very limited. More experiments are needed to convincingly show that GNNExplainer are able to actually focus on most important features. - How does the method perform with changing restrictions on subgraph size, feature dimensions etc.? One can assume the trade-off between larger vs useful explanation but it is important to see this tradeoff by experiments to understand the effectiveness of GNNExplainer in capturing the optimal subcomponents/ sub features that leads to predictions.

Reviewer 4



This paper proposes to provide a model-agnostic approach for providing interpretable explanations for predictions of any GNN-based model by providing an explanation in the form of a small subgraph of the input graph together with a small subset of the most influential node features. The authors validate their proposed approach through several experiments This paper reads well and the results appear sound. I personally find the methodology and the provided techniques to overcome the challenges in each step very interesting. Furthermore, the provided experiments support their intuition and arguments. As for the drawbacks of this paper, I find the relationship to the prior works partly unclear. Furthermore, the related work study is missing some of the most recent works on explaining the learning methods on graphs [1,2,3]. Moreover, It would be nice if the authors could also provide some ideas for future research directions, such as the prospects of using their approach for improving node classification and link prediction tasks. [1] Dai, Hanjun, et al. "Adversarial attack on graph structured data.". [2] Zügner, Daniel, Amir Akbarnejad, and Stephan Günnemann. "Adversarial attacks on neural networks for graph data.". [3] Pezeshkpour, Pouya, Yifan Tian, and Sameer Singh. "Investigating Robustness and Interpretability of Link Prediction via Adversarial Modifications.".

[Author Response · NeurIPS 2019]

We thank the reviewers for their time, valuable feedback, and recommendations for improvements. Overall, the
reviewers found our methodology interesting, novel and technically sound, and our contributions to be very timely.
However, a key point of clarification was raised regarding the selection of hyperparameters and the effects of subgraph
constraints on generated explanations. We address this key point of clarification in detail below. The four reviewers also
raised important clarification points on the motivation for the use of mutual information (R5), synthetic datasets (R2,
R3, R5) and quantitative experiments (R3, R5, R6), and we provide clarification on these issues below as well. These
clarification issues arise—in large part—because certain details were omitted from the paper due to space constraints.
However, an extra page and lengthened appendix will allow us to address these clarification points in the revised version.

**Parameters & effects of constraints on explanations**. R2, R3 and R5 rightly point out the need for more
investigation into hyperparameters and regularization constraints. In new
experiments, we observe that varying regularization strength of constraints
can affect explanations (**Fig. S1**). However, this gives GNNEXPLAINER
flexibility to encode domain-specific priors into constraints and, crucially,
allows the explainer to balance verbosity and completeness of the explana-
tion. In particular, without regularization, the explainer will include many
edges in the explanation even though those edges only negligibly increase
confidence of a GNN's prediction. Conversely, imposing a very small size
constraint $K_M$ will produce meaningless explanations, *e.g.,* single edges.
In practice, we observe that prior knowledge about a task (*e.g.,* select $K_M$
to be the size of a chemical functional group) or commonly used heuris-
tics (*e.g.,* select $K_M$ that results in largest increase of GNN's confidence
score) determine an appropriate value for $K_M$. Furthermore, the range of
acceptable explanation sizes is also important in determining the threshold
for cutting off low-importance edges (*e.g.* how much time a user has to
examine an explanation). In all experiments, we ensure that the setup is
fair for all methods and use the same value for explanation size.

**Figure S1:** Effects of constraint $K_M$ on explanations. On Reddit, we find that modifying $K_M$ doesn't have strong effects, one can easily see the explanatory structure. Effects are stronger when these structures are of fixed size (*e.g.,* 3x3 grid) rather than general shapes (*e.g.,* star or cycle of any size). Further, on Tree-Grid dataset, GNNEXPLAINER is robust to selection of hyperparameter $K_M$; AUCs for $K_M = 6, 8, 10, 12, 14, 16$ are 0.71, 0.83, 0.88, 0.89, 0.86, 0.81, respectively.

**Mutual information**. R5 raises a key point about motivation behind using mutual information. This is indeed
fundamental, but there is intuition for using mutual information: The goal is to identify a small subgraph together
with a small subset of node features that maximize confidence of a GNN's prediction. Mutual information finds
such a subgraph as a connected subgraph of GNN's computation graph. Here, the restriction to be "connected" is
on computation graph rather than the original graph, which is intuitive. Even GNNs that capture information from
far-away nodes through message passing (*e.g.,* Deep Graph Infomax, Jumping Knowledge Networks) have computation
graphs that are different (*i.e.,* multi-hop neighborhoods) but are always connected even though corresponding nodes in
the original graph are not necessarily connected. Further, unlike counterfactual reasoning, mutual information-based
objective allows users to understand what graph structure is critical for a particular prediction in a way that gives a
concise, semantically meaningful explanation. For instance, in Question-Answer graphs of Reddit threads (**Fig. S1**)
we see that explanations have *"2-3 high degree nodes that simultaneously connect to many low degree nodes,"* which
translates to the explanation of *"2-3 experts who all answer many different questions in a QA thread on Reddit."*

**Synthetic datasets**. R3 and R5 raise an important clarification point regarding the description of synthetic datasets.
This is important as these datasets allow us to quantify the quality of explanations without necessitating manual curation,
and thus they represent an advance over the prior art on explaining predictions, which often only provide hand-picked
real-world examples. We acknowledge that the data generation procedure was not adequately explained in the draft and
we will include more information on datasets/groundtruth in the main text with details in the Appendix. Further, we
will open-source the code for data generation together with GNNEXPLAINER's code. Briefly, we first generate a base
graph (*e.g.,* Barabási-Albert graph). For a random set of nodes, we then attach a particular structure (*e.g.,* a house- or
grid-shaped motif) to each of the nodes. These nodes will have labels that are different others, which gives us a dataset
in which absence/presence of the structure indicates a label. The groundtruth thus corresponds to these structures, and
we calculate precision/recall of edges in an explanation relative to edges in the groundtruth.

**New experiments**. In response to the constructive feedback we conducted further experiments whose results confirm
our findings and increase our confidence that GNNEXPLAINER is a general approach for explaining GNNs. Among
others, for R5, we used GNNEXPLAINER to explain GNN models that use simple attention mechanisms and we
observed similarly good performance as when explaining GCNs. Second, in addition to explaining node classification
and graph classification, we also tested GNNEXPLAINER on link prediction (for R3). Experiments on a 2D-Grid dataset,
where each grid graph has 20% of random edges removed, show that explanation for a predicted edge are typically
edges forming a 4-cycle with the predicted edge, consistent with groundtruth. Third, R6 alludes to a hypothetical
situation in which explanation is predictive of a wrong label with high probability. This is very interesting and related to
adversarial attacks, however, we note that this situation was never realized in any of our experiments. That is because, at
GNNEXPLAINER's initialization, the probability of GNN's predicted label is already the highest and thus it is unlikely
that explainer would optimize the subgraph for false prediction. We will include all new analyses in Appendix.

Finally, reviewers raise minor points, such as a typo "$K_M$ nodes" by R5 (it should be $K_M$ edges) and a suggestion
by R6 to investigate how explanations look if their size is constrained by the number of nodes instead of edges (our
experiments show such constraints result in explanations being fully induced subgraphs). We will add formal discussions
to the Appendix, including explicit discussion on future directions and adversarial attacks.

[Meta-Review · NeurIPS 2019]

The reviewers agreed that this paper presents a valuable contribution for explaining GNNs; they appreciated the quality of the writing, the overall motivation of interpretability of the models, and the strength of the empirical results. The primary remaining shortcomings that the reviewers mentioned in the reviews should be addressed as described in the response, such as expanding explanations that describe multiple edges, the significance of the hyper-parameters, description of synthetic datasets, and the additional experiments.